# Geo-Temporal Variation in the Antimicrobial Resistance of *Escherichia coli* in the Community

**DOI:** 10.3390/antibiotics14030233

**Published:** 2025-02-25

**Authors:** Chloé C. H. Smit, Caitlin Keighley, Kris Rogers, Spiros Miyakis, Katja Taxis, Martina Sanderson-Smith, Nick Nicholas, Hamish Robertson, Lisa G. Pont

**Affiliations:** 1Graduate School of Health, University of Technology Sydney, 100 Broadway, Sydney, NSW 2008, Australia; chloe.smit@uts.edu.au (C.C.H.S.); kris.rogers@uts.edu.au (K.R.);; 2Graduate School of Medicine, University of Wollongong, Building 28, Wollongong, NSW 2522, Australia; caitlin.keighley@southernpath.com.au; 3Southern.IML Pathology, 3 Bridge St., Coniston, NSW 2500, Australia; 4Department of Infectious Diseases, Lawson House, Wollongong Hospital, Loftus Street, Wollongong, NSW 2500, Australia; spiros.miyakis@health.nsw.gov.au; 5PharmacoTherapy, -Epidemiology and -Economics, Groningen Research Institute of Pharmacy, University of Groningen, Antonius Deusinglaan 1, 9713 AV Groningen, The Netherlands; k.taxis@rug.nl; 6School of Chemistry and Molecular Bioscience and Molecular Horizons, University of Wollongong, Building 28, Wollongong, NSW 2522, Australia; martina@uow.edu.au; 7School of Design, Queensland University of Technology, 149 Victoria Park Road, Kelvin Grove, QLD 4059, Australia; h5.robertson@qut.edu.au

**Keywords:** antimicrobial resistance, *Escherichia coli*, mapping, geospatial analysis, temporal trends

## Abstract

**Background:** Antimicrobial resistance (AMR) is a global health challenge with significant global variation. Little is known about the prevalence on a smaller geographical scale. **Objectives**: This study aimed to explore the geo-temporal variation in antibiotic resistance in *Escherichia coli (E. coli*) urinary isolates in the Illawarra Shoalhaven region, a region south of Sydney. **Methods**: Data from urine *E. coli* isolates from people living in the community were geospatially analysed from 2008 to 2018. The proportion of resistant isolates was mapped by antibiotic type (amoxicillin with clavulanic acid, cefalexin, norfloxacin, and trimethoprim), postcode, and year. **Results**: Resistance varied by antibiotic, postcode, and over time, with some postcodes showing increased resistance one year and a decrease the following year. Areas with consistently higher resistance included metropolitan, port, and lake regions. We found low resistance in *E. coli* to amoxicillin with clavulanate, cefalexin, and norfloxacin (<5% to 10–19%) and the highest resistance for trimethoprim (10–19% to 30–39%). Overall, from 2008 to 2018, *E. coli* resistance to all four antibiotics increased in this region. **Conclusions**: This study shows temporal and geospatial changes in *E. coli* AMR over small geospatial areas, indicating the opportunity for geospatial analysis to assist in area-specific empirical treatment guidance.

## 1. Introduction

Antimicrobial resistance (AMR) is a global health challenge, in which the One Health perspective is of increasing importance. The One Health perspective considers the environmental, agricultural, and human contexts as a triad, with each contributing individually and in collaboration to increasing AMR development. Taking a One Health approach to AMR allows us to understand, address, and mitigate AMR at a global level [1,2,3]. AMR is primarily driven by natural selection, with the widespread use of antibiotics in both human medicine and agriculture significantly accelerating the rates of AMR [4,5]. The prevalence of AMR varies across the globe [6]. The reasons for these geographical variations are complex, but drivers are likely to include differences in human and animal antimicrobial consumption, antimicrobial policies and stewardship, temperature differences [7], and differences in environmental contamination [8,9].

The exposure of bacteria to antibiotics in the environment increases the chance of the development and spread of resistance [10,11]. The mechanism of resistance varies by antibiotic type, with the acquisition of genes coding for extended-spectrum β-lactamases responsible for resistance to cefalexin and amoxicillin with clavulanic acid, whereas TEM-1/TEM-2- β-lactamase leads to the frequently occurring resistance to amoxicillin [12]. Mutations in the *sul* and *dfr* genes are responsible for resistance to trimethoprim, and mutations involving gyrA/parC and gyrB/parE (responsible for topoisomerase II/IV) lead to resistance to quinolones [12].

In the community, urinary tract infections (UTIs) are a commonly occurring infection for which antibiotics are the primary treatment worldwide. *Escherichia coli (E. coli)* is one of the key pathogens causing community-acquired UTIs in both men and women, responsible for 50–80% of all UTIs [13,14,15]. *E. coli* is a commensal of the human and animal guts and readily exchanges genetic elements, contributing to the spread of resistance [16,17]. Resistance rates of *E. coli* isolates have been shown to vary over time [18,19,20] and geography [6,21,22,23], indicating the need for ongoing AMR surveillance considering both geographical and temporal contexts [23,24,25].

Geo-temporal analysis offers the unique opportunity to explore AMR across geographical and temporal contexts [26]. Changes in AMR over time and between geographical locations are well documented and underpin the use of antibiograms to guide antibiotic treatment [4,27]. While geo-temporal analyses represent a relatively novel approach to the exploration of AMR, they have been used to assist disease pattern detection and hotspot identification during infectious disease outbreaks such as the COVID-19 pandemic and underpin many of the foundations of the modern epidemiology of infectious disease [26,28]. In the context of AMR, this allows the exploration of the spread and the dynamics of resistance while considering geographical influence, aligning with the One Health approach [29]. Geo-temporal analyses have been used to explore trends in *E. coli* resistance in tropical rural Northern Australian, reporting significant geo-temporal variation in AMR [23]. Understanding the temporal patterns in *E. coli* AMR in an urban geospatial location is important in identifying potential hotspot areas that could assist in targeted community (empirical) antibiotic stewardship [23,30].

To date, much of the research into variation in *E. coli* resistance has been conducted in the hospital setting or has looked at variation across large geographical regions or extended aggregate time periods [6,21,23]. Amongst the studies that have explored variation in *E. coli* resistance in the community setting, resistance to a limited number of antibiotics has been explored, many of which do not necessarily represent those commonly in use for managing urinary tract infection (UTI) [31,32,33,34,35]. Therefore, this study aimed to explore local geo-temporal variation in *E. coli* resistance in urine isolates from community-dwelling humans to antibiotics commonly used to treat UTI.

## 2. Results

### 2.1. Rates and Trends of AMR in E. coli

A total of 61,662 urine samples had *E. coli* isolated from 2008 to 2018 in the Illawarra Shoalhaven region. Of these, 31.1% (n = 19,168) were resistant to at least one of the four antibiotics included in our analysis. Resistance was highest for trimethoprim (19.0% of samples, n = 11,729), followed by amoxicillin with clavulanic acid (4.4%, n = 2728) and norfloxacin (4.4%, n = 2470). Resistance to cephalexin was the lowest (3.6%, n = 2241). The number of samples tested for norfloxacin resistance was lower (n = 56,681) due to a global shortage in testing discs between August 2017 and April 2018. For all antibiotics, the rates of resistance increased over the study period (Figure 1).

*E. coli* resistance to trimethoprim increased over time, from 15.1% (95% CI 13.1–17.1) in the last quarter of 2008 to 22.5% (95% CI 20.6–24.4) in the last quarter of 2018 (Figure 1). A similar increase in resistance was observed for amoxicillin clavulanic acid (1.6% (95% CI 0.9–2.3) at the beginning of the study period to 7.4% (95% CI 6.2–8.5) in 2018) and cephalexin (1.5% (95% CI 0.8–2.1) in 2008 to 5.6% (95% CI 4.6–6.6) in 2018). The increase in resistance to norfloxacin was smaller compared to that in amoxicillin and cephalexin (2.1% (95% CI 1.3–2.9) in 2008 to 6.8% (95% CI 5.7–8.0) in 2018).

### 2.2. Geographic Mapping of AMR in E. coli

Figure 2 and Figure 3 show the geo-temporal patterns of *E. coli* isolates resistant to amoxicillin with clavulanic acid, cefalexin, norfloxacin, and trimethoprim per postcode from 2008 to 2012 and from 2013 to 2018 (excluding 2014). Dynamic geo-temporal patterns for 2008–2018 in video format can be found in Appendix A. There was substantial variation in antibiotic resistance between the different postcodes and over the years.

*E. coli* resistance to amoxicillin with clavulanic acid and cefalexin between 2008 and 2012 showed low resistance percentages for all postcodes. Southern postcodes showed variation in resistance in 2013 and 2015 for amoxicillin with clavulanic acid, cefalexin, and norfloxacin, with resistance decreasing to <5% in 2016 for the same antibiotics. Northern postcodes that were found around the metropolitan, port, and lakeside areas showed the highest rates of *E. coli* resistance over the years, varying from 10% to 29% (Table 1). Consistent higher levels of *E. coli* resistance were also noted in two postcodes comprising national parks, which had the lowest number of inhabitants.

## 3. Discussion

We found geo-temporal variation in *E. coli* resistance across local geographies and time frames. Our results indicate that *E. coli* resistance in a single postcode fluctuates with time, highlighting the dynamic nature of community AMR. In our study, postcodes with metropolitan, port, and lake areas were the most likely to maintain consistent rates of *E. coli* resistance. In addition to small geographical variation in *E. coli* resistance, we found that *E. coli* resistance to amoxicillin with clavulanic acid, cefalexin, norfloxacin, and trimethoprim increased over time.

Our results showed that geo-temporal variation in the community setting occurred over time and between postcodes. Our findings are consistent with other community studies investigating local temporal and geospatial variation in *E. coli* resistance [23,32]. However, previous studies have compared larger time frames, for example, comparing 2012–2013 with 2016–2017, and have focused on Extended-Spectrum Beta-lactamase (ESBL) *E. coli*, thus making comparison with the current study difficult [32]. Resistance rates to *E. coli* from the community setting in the Northern Territory in Australia showed a similar dynamic occurrence of resistance rates over time in one study [23], but the higher reported resistance rates have been attributed to remoteness, limited access to health services, a high burden of chronic disease, and complex socio-demographic factors [35].

Interestingly, against a background of increasing resistance, we observed a temporal decrease in resistant isolates for specific antibiotics and postcodes. Some studies indicate that maintaining resistance depends on the fitness cost that is required for it, with bacteria reverting limited energy for reproduction to maintain antibiotic-resistant traits [36]. This is in contrast with another study claiming a lack of detectable fitness cost in reverting resistance to trimethoprim once established [37]. The underlying mechanisms behind *E. coli* acquiring and maintaining resistance need to be further explored from the One Health perspective, highlighting the impact of environmental conditions on the fitness effect in *E. coli* AMR [38].

We observed geospatial variation on a small local scale, with considerable variation noted at the individual postcode level. Our findings of *E. coli* resistance aligning with urban areas and waterways are in line with the previous literature [8,31,33,34] and higher contagion, i.e., person-to-person transmission, of resistant *E. coli* in urban areas [39] and persistence of AMR *E. coli* in water bodies [9,33] have been proposed as the underlying mechanisms behind this. The higher prevalence of *E. coli* resistance observed in lakeside postcodes in our study could indicate transmission between humans and the aquatic environment. Similar findings have been observed in the same region for ESBL-*E. coli* in wastewater [8].

The overall changes in rates of resistance noted in our study were relatively small (<10%), and it is difficult to determine the clinical implications of the variation found in this study since, for most antibiotics, a clinically relevant threshold in resistance has not been established [13]. International guidelines do suggest guidance for trimethoprim and fluoroquinolones, recommending that trimethoprim should not be used for empirical treatment when local resistance exceeds 20% and fluoroquinolones not used when resistance exceeds 10% [13]. Current treatment guidelines in Australia propose trimethoprim as a first-line option for treating uncomplicated UTI [40]; however, our finding that approximately one in five *E. coli* isolates is resistant to trimethoprim highlights the possible need for a revision to the current empirical treatment for UTI in the community setting.

To the best of our knowledge, this is the first study to obtain a comprehensive and inclusive geo-temporal picture of AMR trends in the community at the postcode level in an urban Australian setting. However, like all research, this study has several limitations. An important limitation relevant to the clinical interpretation of our results is the use of aggregate microbiological data. Information on patient characteristics (e.g., sex, age), comorbidities, previous or recurrent antibiotic use, and other information (i.e., urinary devices) may provide additional insights into the likelihood of multi-resistant infections and risk factors, and future research on development of resistance using individual patient data is needed. Another limitation is that we used real-world data with the decision to test left to the treating clinician, and therefore the resistance rates observed in this study may be over-representative of those in the general community, as clinicians may be more likely to test patients for whom they consider that resistance is likely to be an issue. Previous research using the same dataset over a similar time indicated that urine isolates were sampled more frequently in community-dwelling women (88.5%) compared to men (11.6%) [41]. The same research reported that sampling was similar between older (≥65 years, 47.7%) and younger women (<65 years, 52.3%), while for males, sampling was slightly more common among older men (64.1%). Research from a 1998 study suggests that 43.9% of patients presenting with UTI symptoms at the GP received a urine test to guide treatment; however, it is not known whether the testing trends from 1998 are relevant to our study period [42]. Lastly, the study period in this study included the transition by the microbiology provider from the Calibrated Dichotomous Susceptibility (CDS) testing method to the European Committee on Antimicrobial Susceptibility Testing (EUCAST) method. This transition and the use of the different antimicrobial susceptibility testing methods could result in differences in interpretive breakpoints, leading to variation in the resistance rates obtained. At the level of the individual antibiotics, the EUCAST resistance breakpoints were more stringent for trimethoprim and norfloxacin, relatively unchanged for cephalexin, and less stringent for amoxicillin with clavulanic acid; however, the trends observed in our data before and after the change in antibiotic susceptibility methods showed consistently increasing rates in resistance, suggesting that the implications of the findings remain robust.

## 4. Materials and Methods

### 4.1. Setting and Population

The Illawarra Shoalhaven region in New South Wales, Australia, is south of Sydney and covers a geographical area of about 5687 km^2^ (Figure 4) [43,44]. The total population was estimated at 393,204 in 2016 [43] and is expected to increase to 470,000 by 2031 [44]. Microbiology data from this specific region in Australia provide a unique opportunity to explore geospatial variation in AMR since, unlike much of the country, 90% of the community’s microbiology is provided by a single laboratory provider, providing longitudinal microbiology data for the majority of the community-dwelling population [45]. There are 24 postcodes in the region, and there is significant geographical variation between postcodes in terms of geography, environment, and agriculture [46]. Hence, this region provides both longitudinally and geographically varied microbiological data that can be used to understand variation in AMR.

The categorization of geospatial features used to describe the different areas derived from the ABS information (2021) is shown in Table 1 [44].

### 4.2. Data Collection

The dataset used in this study comprised de-identified community-dwelling microbiology data from routine urine culture requests, processed using standard microbiological procedures. Only the first isolate per patient was included. Details on the dataset and susceptibility testing used to produce this dataset have been described elsewhere [41,45]. Urine samples were inoculated onto blood agar, MacConkey agar, and CHROMagar media and incubated at 37 degrees for 24–48 h as per local laboratory protocols [41]. For cultures to be defined as positive for *E. coli*, they needed to have a bacterial count of ≥10^7^ colony-forming units (CFUs) per millilitre [47,48,49]. Consistent with previous work using this dataset, data from November 2013 to January 2015 were excluded due to a changeover in the susceptibility testing method. Before 30 November 2013, the laboratory provider used agar disc diffusion via the Calibrated Dichotomous Susceptibility (CDS) method. From 2015 onwards, the European Committee on Antimicrobial Susceptibility Testing (EUCAST) disc diffusion method was used (version 5.0, 2015 to version 8.0, 2018). With the CDS method, isolates were categorised either as resistant (R) or susceptible (S), whereas isolates were categorised as resistant (R), susceptible (S), or intermediate (I) with the EUCAST method used over this period [50]. As per previous work using this dataset, isolates in the intermediate category were grouped with resistant isolates [45]. The differences in clinical breakpoints for the two methods and after re-categorization are compared in Table 2.

The dataset comprised aggregate data, including the date (year and month), antibiotic type, number of isolates resistant/susceptible to the antibiotic, and patient residential postcodes.

In Australia, antibiotics included in the microbiology antibiotic susceptibility testing panel may vary between laboratories, with each laboratory identifying the antibiotics relevant to their area based on current treatment guidelines and local clinical practice [51]. Current first-line options in the Australian guidelines specify trimethoprim (as monotherapy and therefore not in combination with sulfamethoxazole), nitrofurantoin, and cephalexin as first-line treatment options [40]. The microbiological antibiotic susceptibility testing panel for E. coli urine isolates used in the Illawarra Shoalhaven region includes testing for resistance to amoxicillin, amoxicillin with clavulanic acid, cefalexin, nitrofurantoin, norfloxacin, and trimethoprim, reflecting current first-line treatment recommendations in the Australian prescribing guidelines for the management of urinary tract infection [52]. Nitrofurantoin was excluded from our analyses due to incomplete microbiology data collection [41]. For this analysis, we excluded amoxicillin since it is no longer recommended for use in the treatment of E. coli due to poor efficacy and high resistance, with around 50% of isolates reported as being resistant [13].

### 4.3. Statistical Analysis

We used descriptive statistics to calculate the proportion of *E. coli* isolate resistant to each antibiotic per antibiotic, per postcode, and per quarter (three months). The proportion of resistance was calculated as the total number of isolates resistant to the antibiotic divided by the total number of isolates tested in one quarter in one postcode.

The overall *E. coli* resistance rates per antibiotic per quarter were plotted in a scatterplot with 95% confidence intervals. The aggregated yearly *E. coli* resistance rates per antibiotic and postcode were categorised as <5%, 5–9%, 10–19%, 20–29%, and 30–39% and mapped.

### 4.4. Geocoding and Mapping

We used R studio statistical package R 1.4.1106 (R Foundation for Statistical Computing, Vienna, Austria) to create, edit, and analyse the geospatial AMR information. Within this software, we used the ggplot2 (version 3.5.0) and the sf package (version 1.0-1.5), which required coordinates to map the data. Using a spatial join, we linked the coordinate information from the ABS website to the postcodes in our data. Colour-themed maps (yellow, light orange, orange, red, and deep red) were created to visually display the categories for the proportion of *E. coli* AMR per antibiotic type and postcode.

## 5. Conclusions

In this research, we showed considerable variation in *E. coli* resistance across a much smaller geographical scale than previously reported and showed the importance of including variation across time and geography in any analysis of AMR trends. Geo-temporal analysis provides valuable insights for understanding changes in AMR and guides the development of locally relevant treatment choices. In our work, we showed increasing resistance of *E. coli* to trimethoprim, suggesting the need to review current empirical treatment in the community in Australia. Future geo-temporal research should include clinical information from patients to confirm these findings and to guide antibiotic prescribing in the community.

## Figures and Tables

**Figure 1 antibiotics-14-00233-f001:**
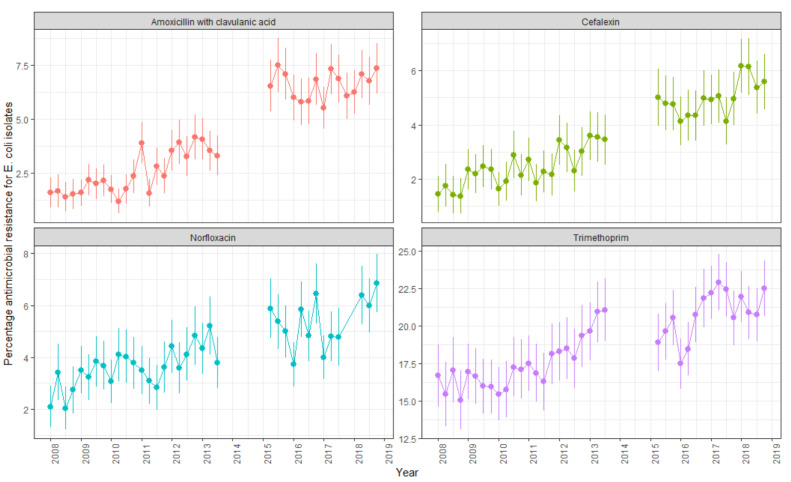
Quarterly rates of *E. coli* AMR ± 95% confidence intervals from urine isolates obtained from people in the Illawarra Shoalhaven community per antibiotic type. The gap in data in 2014 is due to a change in the susceptibility testing method from agar disc diffusion via the Calibrated Dichotomous Susceptibility (CDS) method (used 2008–2014) to the European Committee on Antimicrobial Susceptibility Testing (EUCAST) disc diffusion method (from 2015 onwards).

**Figure 2 antibiotics-14-00233-f002:**
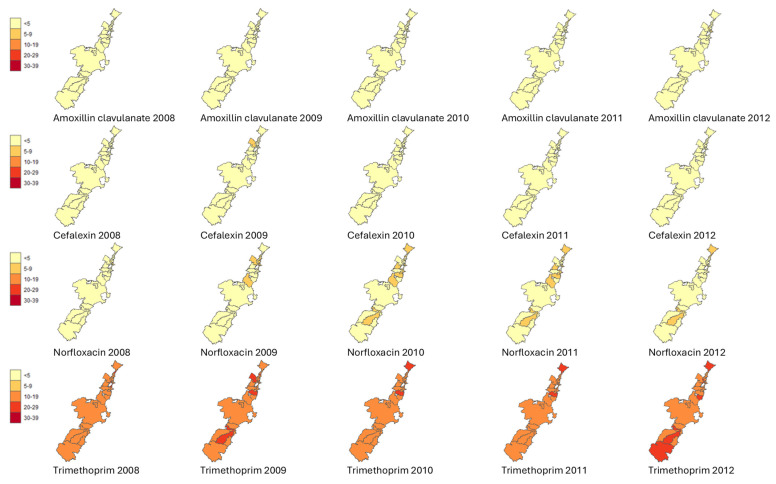
Visualisation of the geo-temporal rates of *E. coli* resistance per postcode in the Illawarra Shoalhaven region from 2008 to 2012. Light yellow, <5%; light orange, 5–9%; dark orange, 10–19%; red, 20–29%; and dark red, 30–39% of *E. coli* isolates were resistant per antibiotic type.

**Figure 3 antibiotics-14-00233-f003:**
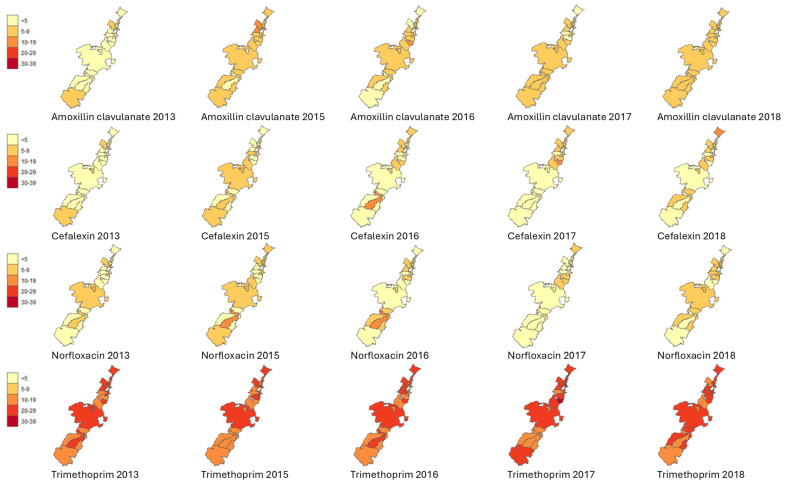
Visualisation of the geo-temporal rates of *E. coli* resistance per postcode in the Illawarra Shoalhaven region from 2013 to 2018 (excluding 2014). Light yellow, <5%; light orange, 5–9%; dark orange, 10–19%; red, 20–29%; and dark red, 30–39% of *E. coli* isolates were resistant per antibiotic type.

**Figure 4 antibiotics-14-00233-f004:**
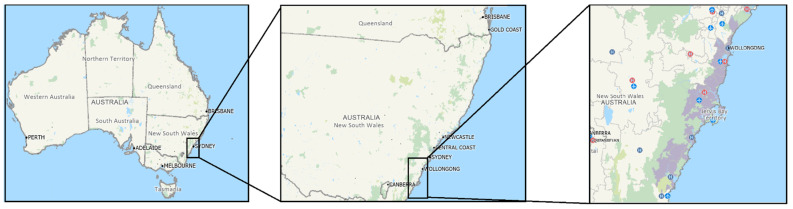
The Illawarra Shoalhaven region has population-dense regions (purple), hospitals (red H), airports (aviation sign), the ocean (blue), and national parks (green). We created these maps using Maptitude software (Caliper Corporation, 2023, Newton, MA, USA, www.MappingSoftware.com, access date: 10 December 2024).

**Table 1 antibiotics-14-00233-t001:** Categorization of the 24 postcodes in the Illawarra Shoalhaven region into rural, urban, lake, and port areas.

Location Category	Postcode
Rural	2508, 2527, 2533, 2534, 2535, 2536, 2538, 2539, 2540
Urban/Metropolitan	2500, 2515, 2516, 2517, 2818, 2519, 2525, 2526, 2528, 2541
Lake Illawarra	2506, 2528, 2529, 2530
Port	2502, 2505

**Table 2 antibiotics-14-00233-t002:** Clinical breakpoints for CDS and EUCAST for antibiotics included in the susceptibility testing panel for Enterobacteriaceae (*Escherichia coli)* urine isolates. The breakpoints remained the same for both methods across the years.

Antibiotic	MIC Breakpoint CDS (2008–2013) (mg/L)	Breakpoint EUCAST (2015–2018) (mg/L) Version 5.0, 2015 to Version 8.0, 2018
Amoxicillin with clavulanic acid	Susceptible (S): ≤8/4 Resistant (R): >8/4	Susceptible (S): ≤32/2 Resistant (R): >32/2
Cefalexin	Susceptible (S): ≤16 Resistant (R): >16	Susceptible (S): ≤16 Resistant (R): >16
Norfloxacin	Susceptible (S): ≤4 Resistant (R): >4	Susceptible (S): ≤0.5 Resistant (R): >1
Trimethoprim	Susceptible (S): ≤4 Resistant (R): >4	Susceptible (S): ≤2 Resistant (R): >4

## Data Availability

Restrictions apply to the availability of data used in this study. Data were obtained from Southern IML Pathology/WARRA and are available from the authors with permission from these providers.

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
