# Peer review of "Geo-Temporal Variation in the Antimicrobial Resistance of *Escherichia coli* in the Community"

_antibiotics, 2025, doi:10.3390/antibiotics14030233_

Round 1

Reviewer 1 Report

Comments and Suggestions for Authors

The manuscript addresses a highly relevant and timely issue in public health: the geo-temporal variation in antimicrobial resistance (AMR) of Escherichia coli in the community. The study’s focus on small geographical scales and its integration of spatiotemporal mapping are valuable contributions to understanding AMR dynamics in the community setting. The methodology is robust, utilizing a large dataset over a 10-year period, and the visual mapping enhances the interpretability of the findings. In general the manuscript is well-written. However, there are several areas for improvement and clarification to strengthen the manuscript.

Some of my suggestions for improvement:

1. Expand on how geo-temporal studies can offer unique insights compared to traditional spatial or temporal analyses. For example, discuss how this approach could reveal trends that might be masked in broader studies. Include examples or references to prior studies where geo-temporal analysis led to actionable public health outcomes in Introduction. 

2. Explicitly state why the Illawarra Shoalhaven region is a suitable case study for geo-temporal analysis in Introduction for refine transition to the objective of study.

3. The gap in data during the transition from CDS to EUCAST susceptibility testing methods may introduce inconsistencies in the temporal analysis. Please explain the potential impact of differing resistance classification criteria on the results.

4. Please add conclusion section to summarize the key takeaways or their broader implications. Summarizing the primary outcomes of the study, such as the observed geo-temporal trends in E. coli AMR and the identification of resistance hotspots, and its implication for policies towards antimicrobials used. Also add the limitation and suggestion that will be useful for further research.

Author Response

The manuscript addresses a highly relevant and timely issue in public health: the geo-temporal variation in antimicrobial resistance (AMR) of Escherichia coli in the community. The study’s focus on small geographical scales and its integration of spatiotemporal mapping are valuable contributions to understanding AMR dynamics in the community setting. The methodology is robust, utilizing a large dataset over a 10-year period, and the visual mapping enhances the interpretability of the findings. In general, the manuscript is well-written. However, there are several areas for improvement and clarification to strengthen the manuscript.

Some of my suggestions for improvement:

  1. Expand on how geo-temporal studies can offer unique insights compared to traditional spatial or temporal analyses. For example, discuss how this approach could reveal trends that might be masked in broader studies. Include examples or references to prior studies where geo-temporal analysis led to actionable public health outcomes in Introduction. 

Response 1: Thank you for this useful comment. We have revised the introduction to articulate the unique insights that geo-temporal analyses can offer including examples of where geo-temporal analyses have contributed to improved public health.

  1. Explicitly state why the Illawarra Shoalhaven region is a suitable case study for geo-temporal analysis in Introduction for refine transition to the objective of study.

Response 2: Thank you for this comment that strengthens our manuscript. The description of the Illawarra region in the Methods section has been expanded to include deeper consideration of why this location provided unique opportunities as a case study for geo-temporal analyses of AMR. 

  1. The gap in data during the transition from CDS to EUCAST susceptibility testing methods may introduce inconsistencies in the temporal analysis. Please explain the potential impact of differing resistance classification criteria on the results.

Response 3: To allow the reader to better understand the implications of the change from CDS to EUCAST susceptibility testing we have added the different MIC breakpoints for both methods used in the method section of the manuscript and included the limitations and implications of these in the discussion section.

  1. Please add conclusion section to summarize the key takeaways or their broader implications. Summarizing the primary outcomes of the study, such as the observed geo-temporal trends in E. coli AMR and the identification of resistance hotspots, and its implication for policies towards antimicrobials used. Also add the limitation and suggestion that will be useful for further research.

Response 4: Thank you for these very clear and useful suggestions. We have added a conclusion section at the end of the discussion summarizing the key takeaways and their broader implications.

Reviewer 2 Report

Comments and Suggestions for Authors

This is an interesting work focusing on the study of geo-temporal evolution of E coli resistance against specific antibiotic in Illawarra Shoalhaven, a region south of Sydney. However, I have some substantial remarks on this work which should be resolved:

*1/The choice of the four antibiotics is not clearly justified since for example trimethoprim is generally used in combination with Sulfamethoxazole;

*2/As discussed “Firstly, the lack of patient data is an important limitation of this study for the clinical implications of our findings. Information on patient characteristics (e.g., sex, age), comorbidities, previous or recurrent antibiotic use and other information (i.e., urinary devices) may have provided valuable insight into the likelihood of multi-resistant infections and contributing risk factors”. This is a big limitation for this study;

Minor points:

·         *The introduction should be enriched with a paragraph focusing on the molecular aspects supporting resistance to antibiotics

·         *The title and legend of figure 3 are out of place

·      *  The medium used for isolation and identification of E. coli should be added. If there is any standard used, this should be mentioned in the text.

Author Response

This is an interesting work focusing on the study of geo-temporal evolution of E coli resistance against specific antibiotic in Illawarra Shoalhaven, a region south of Sydney. However, I have some substantial remarks on this work which should be resolved:

*1/The choice of the four antibiotics is not clearly justified since for example trimethoprim is generally used in combination with Sulfamethoxazole;

Response 1: Thank you for this useful comment. In Australia, trimethoprim without sulfamethoxazole is first-line therapy for uncomplicated urinary tract infection in our national treatment guidelines and trimethoprim with sulfamethoxazole is not routinely used for the management of urinary tract infection.  We have clarified this in the methods in the section regarding the antibiotic susceptibility testing panel.

*2/As discussed “Firstly, the lack of patient data is an important limitation of this study for the clinical implications of our findings. Information on patient characteristics (e.g., sex, age), comorbidities, previous or recurrent antibiotic use and other information (i.e., urinary devices) may have provided valuable insight into the likelihood of multi-resistant infections and contributing risk factors”. This is a big limitation for this study;

Response 2: We agree that further research using individual patient-level data is needed and as noted have identified and discussed this as a limitation to the study in the discussion section.

Minor points:

  •        *The introduction should be enriched with a paragraph focusing on the molecular aspects supporting resistance to antibiotics

We have revised the introduction to add consideration of the molecular aspects of AMR.

  •        *The title and legend of figure 3 are out of place

Response 4: Thank you for pointing this out. We have corrected this.

  •     *  The medium used for isolation and identification of E. coli should be added. If there is any standard used, this should be mentioned in the text.

Response 5: We have added this to the data collection section of the methods.

Reviewer 3 Report

Comments and Suggestions for Authors

The current manuscript evaluates the geo-temporal variations in AMR for E. coli on a community level in a region of Australia.

Major points

- the data for trimethoprim refer to trimethoprim alone or in combination with Sulfamethoxazole? this fact should be stated more clearly in the text. Trimethoprim alone would be somewhat unusual, as these antibiotics are usually tested together

- Line 54 – while E. coli is certainly the most frequent pathogen of UTIs in women, the same is not always true for men. The text should reflect this fact

- Line 65 – these references seem to refer to other regions of the globe. Are there any previous studies that examine AMR in the currently studied zone of Australia?

- Figure 1 – „Quarterly rates of E. coli AMR ± 95% confidence intervals from urine people in the” – urine people?

- the geo-temporal patterns of AMR for 2008 – 2010 should be added to the main text from the appendix. This would enhance the readability of the Figures and respective text

- “Current treatment guidelines in Australia” – is Fosfomycin not recommended and used empirically in Australia for UTIs determined by E. coli?

- “could be extrapolated to other regions to forecast AMR” – while the overall trend in AMR seems to be increasing worldwide and subsequently also in the examined area, I don’t believe that these data could actually be used to forecast future AMR rates. Expand this idea in a practical manner

- what would be some practical conclusions regarding empirical treatment based on these results?

- Conclusions are missing

- Methods – antibiotic susceptibility testing – the number of tested antimicrobials is quite low in this study. Any particular reasons for this fact?

- Methods - the changes in the antimicrobial susceptibility testing are not entirely clear from the text. Please rephrase this part as to improve clarity. Furthermore add the exact guidelines (with version and year) used for the interpretation of the AST

Minor points

- Figure 3 – the text should be right below the image. The formatting needs to be redone

- “to have a bacterial count of ≥107 colony-forming units (CFU) per liter” – redo with 107 and also add the more commonly used metric of CFU/mL

- some English corrections are required throughout the manuscript

Comments on the Quality of English Language

Some English improvements are required in the manuscript

Author Response

The current manuscript evaluates the geo-temporal variations in AMR for E. coli on a community level in a region of Australia.

Major points

- the data for trimethoprim refer to trimethoprim alone or in combination with Sulfamethoxazole? this fact should be stated more clearly in the text. Trimethoprim alone would be somewhat unusual, as these antibiotics are usually tested together

Response 1:

Thank you for this useful comment. In Australia, not all antimicrobials are tested against all organisms, and laboratories have protocols for testing that align with National treatment guidelines and local clinical practice. Trimethoprim is first-line choice in National Australian treatment guidelines for the management of UTI in the community and therefore included in the routine testing panel. This has been clarified in the methods section.

- Line 54 – while E. coli is certainly the most frequent pathogen of UTIs in women, the same is not always true for men. The text should reflect this fact

Response 2: Thank you for this comment, we have revised this in the introduction based on the work of Bielec et al. (https://doi.org/ 10.3390/jcm12165166), who found that regardless of sex, 51-84% of community-acquired urinary tract infections are caused by Escherichia coli.

- Line 65 – these references seem to refer to other regions of the globe. Are there any previous studies that examine AMR in the currently studied zone of Australia?

Response 3: Thank you for pointing this out. This is the first geo-temporal analyses in the Illawarra Shoalhaven region. With geo-temporal analyses, a relatively novel approach to AMR analyses, to-date limited studies using this approach exist. To the best of our knowledge, only one other geo-temporal analysis of AMR has been conducted in rural Australia (exploring E. coli across much larger geographies in tropical Norther Australia) and this reference has been added to the introduction.   

- Figure 1 – „Quarterly rates of E. coli AMR ± 95% confidence intervals from urine people in the” – urine people?

Response 4: Thank you for this comment. It helped clarify the description of the figure. We have revised this to clarify the text.

- the geo-temporal patterns of AMR for 2008 – 2010 should be added to the main text from the appendix. This would enhance the readability of the Figures and respective text

Response 5: We agree and have combined the years 2008 – 2012 in Figure 2 and the years 2013 to 2018 in Figure 3 to ensure the readability of the maps.

- “Current treatment guidelines in Australia” – is Fosfomycin not recommended and used empirically in Australia for UTIs determined by E. coli?

Response 6: Thank you for this useful comment. Fosfomycin, while a second line therapy for targeted treatment in our national guidelines, is rarely utilized in Australia since it is not reimbursed as part of the National reimbursement scheme. While not in the scope of this study previous research by our group found almost no prescribing of Fosfomycin (one patient only) in the Illawarra Shoalhaven region.  

As discussed above, we have revised the methods to articulate the choice of antibiotics included in the study as well as adding the current treatment guidelines for management of urinary tract infection in Australia

- “could be extrapolated to other regions to forecast AMR” – while the overall trend in AMR seems to be increasing worldwide and subsequently also in the examined area, I don’t believe that these data could actually be used to forecast future AMR rates. Expand this idea in a practical manner

Response 7: Thank you for this comment, we agree that the current analysis could not be used for forecasting AMR however it does demonstrate the needs to consider time and geography in any future forecasting analyses.  This has been revised in the discussion.

- what would be some practical conclusions regarding empirical treatment based on these results?

Response 8: We have added an explicit conclusion at the end of the discussion section.

- Conclusions are missing

Response 9: As noted above, we have added an explicit conclusion at the end of the discussion section.

- Methods – antibiotic susceptibility testing – the number of tested antimicrobials is quite low in this study. Any particular reasons for this fact?

Response 10: The choice of antibiotics tested has been discussed under response 1.

- Methods - the changes in the antimicrobial susceptibility testing are not entirely clear from the text. Please rephrase this part as to improve clarity. Furthermore, add the exact guidelines (with version and year) used for the interpretation of the AST

 Response 11: This has been clarified in the limitation section of the discussion.

Minor points

- Figure 3 – the text should be right below the image. The formatting needs to be redone

Response 12: We have corrected this 

- “to have a bacterial count of ≥107 colony-forming units (CFU) per liter” – redo with 107 and also add the more commonly used metric of CFU/mL

Response 13: Thank you for your suggestion, this has been amended:

- some English corrections are required throughout the manuscript

Response 14: Thank you for this comment. The manuscript has been edited and reviewed by a native English-speaking author.

Round 2

Reviewer 2 Report

Comments and Suggestions for Authors

-

Author Response

We thank the reviewers for their valuable time and help to improve our work. 

Reviewer 3 Report

Comments and Suggestions for Authors

The manuscript was sufficiently improved by the authors, who addressed my concerns and suggestions.

One aspect still requires improvement - line 252: in CFU/ml the correct value is 100.000 meaning (10)5

Author Response

Thank you for your time and critical eye in reviewing our work. It significantly improved the quality of your manuscript.